# Examination of the Daily Cycle Wind Vector Modes of Variability from the Constellation of Microwave Scatterometers and Radiometers

**Francis Joseph Turk** [1,*], **Svetla Hristova-Veleva** [1] and **Donata Giglio** [2]

1   Jet Propulsion Laboratory (JPL), California Institute of Technology, Pasadena, CA 91109, USA;
    svetla.hristova@jpl.nasa.gov
2   Department of Atmospheric and Oceanic Sciences, University of Colorado, Boulder, CO 80302, USA;
    Donata.Giglio@Colorado.EDU
*   Correspondence: jturk@jpl.caltech.edu

**Abstract:** Offshore of many coastal regions, the ocean surface wind varies in speed and direction throughout the day, owing to forcing from land/sea temperature differences and orographic effects. Far offshore, both diurnal and semidiurnal wind vector variability has been noted in the Tropical Atmosphere Ocean-TRIangle Trans-Ocean buoy Network (TAO-TRITON) mooring data in the tropical Pacific Ocean. In this manuscript, the tropical diurnal wind variability is examined with microwave radiometer-derived winds from the Tropical Rainfall Measuring Mission (TRMM) and the Global Precipitation Measurement (GPM), merged with RapidScat and other scatterometer data. Since the relationship between wind speed and its zonal and meridional components is non-linear, this manuscript describes an observationally based methodology to merge the radiometer and scatterometer-based wind estimates as a function of observation time, to generate a multi-year dataset of diurnal wind variability. Compared to TAO-TRITON mooring array data, the merged satellite-derived wind components fairly well replicate the semidiurnal zonal wind variability over the tropical Pacific but generally show more variability in the meridional wind components. The meridional component agrees with the associated mooring location data in some locations better than others, or it shows no clear dominant diurnal or semidiurnal mode. Similar discrepancies are noted between two forecast model reanalysis products. It is hypothesized that the discrepancies amongst the meridional winds are due to interactions between surface convergence and convective precipitation over tropical ocean basins.

**Keywords:** ocean; wind; tropical; diurnal; RapidScat; TRMM; GPM; scatterometer; radiometer; microwave

## 1. Introduction

Over many oceanic regions, the surface wind varies in speed and direction throughout the day, owing to meteorological forcing such as coastal land/sea temperature differences and orographic effects [1–4]. In addition to these near-land effects, diurnal variability in tropical oceanic convective precipitation, far from continents, has been noted from analysis of Tropical Rainfall Measuring Mission (TRMM) data [5–7], indicating an early morning maximum. Over the intertropical convergence zone (ITCZ), mooring data from the Tropical Atmosphere Ocean-TRIangle Trans-Ocean buoy Network (TAO-TRITON) [8] reveal both diurnal and semidiurnal wind vector variability [9,10]. To date, relatively few observational (satellite-based) investigations have been undertaken that attempt to estimate these daily modes of ocean surface wind variability (hereafter, the term daily refers to all variations within a 24-h period, and diurnal and semidiurnal will refer to the first and second harmonics of the daily cycle, respectively). Such an observational capability would be useful to understand the mechanisms linking convective precipitation to the diurnal

cycle of surface wind and wind convergence [11,12]. More specific to linking winds and convective precipitation are the dynamically important quantities of time-mean surface divergence and vorticity, which can be computed from the zonal ($u$) and meridional ($v$) wind components estimated by scatterometers [13,14]. Moreover, coupling oceanic wind modes with independent satellite-based precipitation structure would provide a unique evaluation of climate models. The ability of a climate model to represent the joint daily modes of variability of wind and precipitation over the ocean provides a test of the physical convective parameterizations used in the model [15,16].

Just as the TRMM represented the first opportunity to sample the diurnal variability of tropical precipitation, the deployment of the RapidScat scatterometer onboard the International Space Station (ISS) in 2014 represented the first opportunity to sample the diurnal variability of ocean surface winds [17]. This is due to the non-sun-synchronous nature of the TRMM and ISS orbits, i.e., the observation is asynchronous (i.e., not tied to particular local solar viewing angles, but rather precesses over several months' time). As presented in Table 1, the data record of satellite-based ocean surface wind observations consists of a diverse collection of both sun-synchronous orbiting satellites, such as the long period of Defense Meteorological Satellite (DMSP) platforms, and asynchronous (e.g., RapidScat, TRMM, GPM) orbiting satellite platforms. Other than the Coriolis-WindSat polarimeter [18], wind-speed-only ($w$) measurements are available from the passive microwave (MW) radiometers and wind vectors ($u$, $v$) available from the scatterometers [19].

**Table 1.** Source and time period of satellite-derived ocean winds used in this investigation [1]. Bold font indicates sensors with a windspeed-only capability. DSMP LTAN drift is noted.

| Satellite | Sensor | Source | Posted Resolution (km) | Period Covered | LTAN |
|---|---|---|---|---|---|
| SeaWinds | SeaWinds | RSS | 25 | 04/2003–10/2003 | 2230 |
| QuikSCAT | SeaWinds | RSS | 25 | 04/2003–11/2009 | 0600 |
| QuikSCAT (nonspinning) | SeaWinds | PO.DAAC | 12.5 | 2010–2017 (intermittent) | 0600 |
| ISS | RapidScat | PO.DAAC | 12.5 | 10/2014–09/2016 | variable |
| Coriolis | WindSat | RSS | 25 | 04/2003–02/2017 | 1800 |
| Oceansat-2 | OSCAT | PO.DAAC | 12.5 | 01/2010–02/2014 | 1200 |
| MetOp-A | ASCAT | RSS | 25 | 04/2007–02/2017 | 2130 |
| MetOp-B | ASCAT | RSS | 25 | 11/2012–02/2017 | 2130 |
| **DMSP F-16** | **SSMIS** | RSS | 25 | 01/2004–02/2017 | 2015-1600 |
| **DMSP F-17** | **SSMIS** | RSS | 25 | 12/2006–02/2017 | 1730-1830 |
| **Aqua** | **AMSR-E** | RSS | 25 | 04/2003–10/2011 | 1330 |
| **GCOM-W** | **AMSR-2** | RSS | 25 | 07/2012–02/2017 | 1330 |
| **TRMM** | **TMI** | RSS | 25 | 04/2003–12/2014 | variable |
| **GPM** | **GMI** | RSS | 25 | 03/2014–02/2017 | variable |

[1] DMSP = Defense Meteorological Satellite Program, ISS = International Space Station, GCOM-W = Global Change Observing Mission for Water, TRMM = Tropical Rainfall Measuring Mission, GPM = Global Precipitation Measurement, AMSR-E = Advanced Microwave Scanning Radiometer for EOS, SSMIS = Special Sensor Microwave Imager Sounder, TMI = TRMM Microwave Imager, GMI = GPM Microwave Imager, ASCAT = Advanced Scatterometer, OSCAT = Oceansat-2 Scatterometer, LTAN = Local Time of Ascending Node.

Prior to RapidScat, there had been relatively few satellite-based investigations undertaken that attempted to estimate the daily modes of ocean surface wind variability. The best known of these is the brief seven-month "tandem period" in 2003, overlapping the Seawinds scatterometer onboard the Advanced Earth Observation Satellite (ADEOS-2) and the QuikSCAT satellite, which provided four ($u$, $v$) observations per day, nearly equally spaced in time, enabling estimation of the diurnal mode. These data triggered several studies of the diurnal ocean surface wind vector variability [11,20,21] and timing of oceanic convergence [22]. Owing to the brief period, these data are indicative of the underlying meteorology during that particular year and likely to be insufficient in regions of strong synoptic variability. The 2010–2014 and 2017–current periods provide overlaps between the OceanSat Ku-band scatterometers operated by the Indian Space Research Organization (ISRO) and the C-band Advanced Scatterometer (ASCAT) operated by EUMETSAT, but the

diurnal sampling is from a less optimal local time spacing (0000/1200 local for OceanSat-2, 0930/2230 for the EUMETSAT MetOp platforms). Since 2017, the eight-satellite Cyclone Global Navigation Satellite System (CYGNSS) has been providing ocean surface wind speeds at bistatic specular reflection points, which are traced out along the ocean surface depending upon the locations of the transmitting Global Positioning System (GPS) spacecraft [23]. Unlike GPM, whose (subpoint) local observing time repeats every 46 days at the equator, the time and location of the CYGNSS observations occur randomly due to the asynchronous nature of the CYGNSS and Global Positioning System (GPS) satellite orbits. The capability of CYGNSS and ASCAT to capture the diurnal wind patterns noted in earlier studies was studied by [24]. However, the few methods that have attempted to merge the longer-term record of multiple wind speed and wind vector observations [25,26] are not of a temporal and spatial scale appropriate for both diurnal and semidiurnal wind analysis.

Since the relationship between wind speed and its ($u$, $v$) components is nonlinear, this manuscript describes an observationally based methodology to merge the MW radiometer and scatterometer-based wind estimates as a function of observation time, to generate a multi-year dataset of diurnal wind variability. Section 2 describes the analysis used to derive the diurnal and semidiurnal wind components when a sufficient number of satellite observations are available on any given day. Section 3 compares the results of these analyses to the diurnal wind modes revealed from the TAO-TRITON moorings in the tropical Pacific and a similar analysis from two forecast model reanalyses over the same region. After discussion in Section 4, the manuscript concludes in Section 5 with possible explanations for discrepancies between models and observations.

## 2. Data Sources and Method

In this section, the sources and characteristics of the data are presented. The methodology used to assign a wind direction to the wind-speed-only observations is described.

### 2.1. Satellite, Mooring and Model Data Sources

Sources of satellite-derived winds are listed in Table 1. All MW radiometer-based wind datasets were obtained from the extensive satellite-derived winds archive produced and distributed by Remote Sensing Systems (RSS). After 2009, QuikSCAT data were collected during intermittent periods in a staring (non-spinning) mode to enable calibration of other scatterometers. RapidScat and non-spinning QuikSCAT data were obtained from the NASA Physical Oceanography Distributed Active Archive Center (PO.DAAC). This effort will utilize the recent Level-2 wind products developed for TMI [27], intercalibrated with other MW radiometers [19].

Importantly, the datasets were screened for the presence of rain and intercalibrated to the TMI reference sensor, such that the overall relative bias compared to wind speeds derived from the other individual sensors in Table 1 was less than 0.1 m s$^{-1}$ [27]. Figure 1 shows individual wind speed histograms (derived from all ±15-min coincidences, identical data counts and essentially identical observation times) between TRMM and QuikSCAT during two different years (2003 and 2008), under rain-free conditions. For speeds less than 15 m s$^{-1}$, there is no significant bias. On average, for wind speeds > 15 m s$^{-1}$, QuikSCAT is biased slightly low relative to TMI. It is noted that wind speeds in this high range are two orders of magnitude or fewer in number than the predominant winds in the 5–10 m s$^{-1}$ range and occur more frequently outside of the tropical Pacific Ocean domain (8S-8N) investigated here.

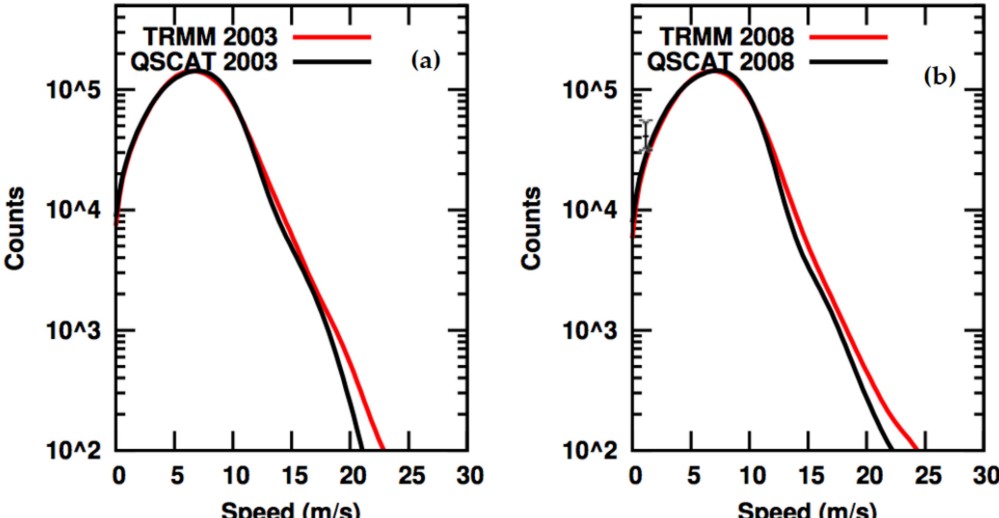

**Figure 1.** Individual wind speed histograms resulting from all 15-min coincidences between TRMM and QuikSCAT during (**a**) 2003 and (**b**) 2008, under rain-free conditions.

Data from the TAO-TRITON global tropical moored buoy array [8] were obtained from the Global Tropical Moored Buoy Array archive at the Pacific Marine Environmental Laboratory (PMEL). TAO-TRITON moorings are located along eight longitudes in the tropical Pacific between 170E and 95W longitude, at seven latitudes (8S, 5S, 2S, 0N, 2N, 5N, 8N) for most moorings. For each day that mooring data were available between 2003 and 2017, the high-quality (10-min sampling) data were time-averaged to hourly and aggregated into 24 local time bins.

Ocean surface wind data from two forecast model reanalyses were examined for comparison with the mooring array daily wind components. These are the ERA-Interim (ERA-I) global atmospheric reanalysis produced by the European Centre for Medium-Range Weather Forecasts (ECMWF) [28] and the Modern-Era Retrospective Analysis for Research and Applications, Version 2 (MERRA2) reanalysis produced by the NASA Global Modeling and Assimilation Office (GMAO) [29]. These data are available at 3-hourly intervals. For each day that these model data are available between 2003 and 2017, the surface winds from the MERRA2 and ERA-I reanalysis grid box closest to the TAO-TRITON mooring array locations are located (8 local time bins per day).

### 2.2. Vector Reconstruction Methodology

Previous studies [20,30] have noted that the daily wind pattern near coastal regions follows a near-elliptical variability. Assume that there are $n$ wind vector estimates from $n$ satellite passes, all of them during a given day and over a given location. This number varies depending upon the time of year, location and the particular satellites being considered.

Figure 2 shows that during 2010 (top panel) near the equator, the sun-synchronous OceanSat-2 (violet symbols) and ASCAT (blue symbols) scatterometers and the WindSat MW polarimeter (orange symbols) were operating, and $n$ could reach values of 6 for some locations on some days, if the location was captured by all three instruments during the ascending and descending portions of their orbits. During 2015 (lower panel), ocean vector observations were available from ASCAT, WindSat and also the asynchronous RapidScat (red symbols). Approximately every 30 days, the RapidScat observing time would repeat, and when its observations occurred near 12–15 local time, they filled in an otherwise lengthy revisit time gap between ASCAT and WindSat.

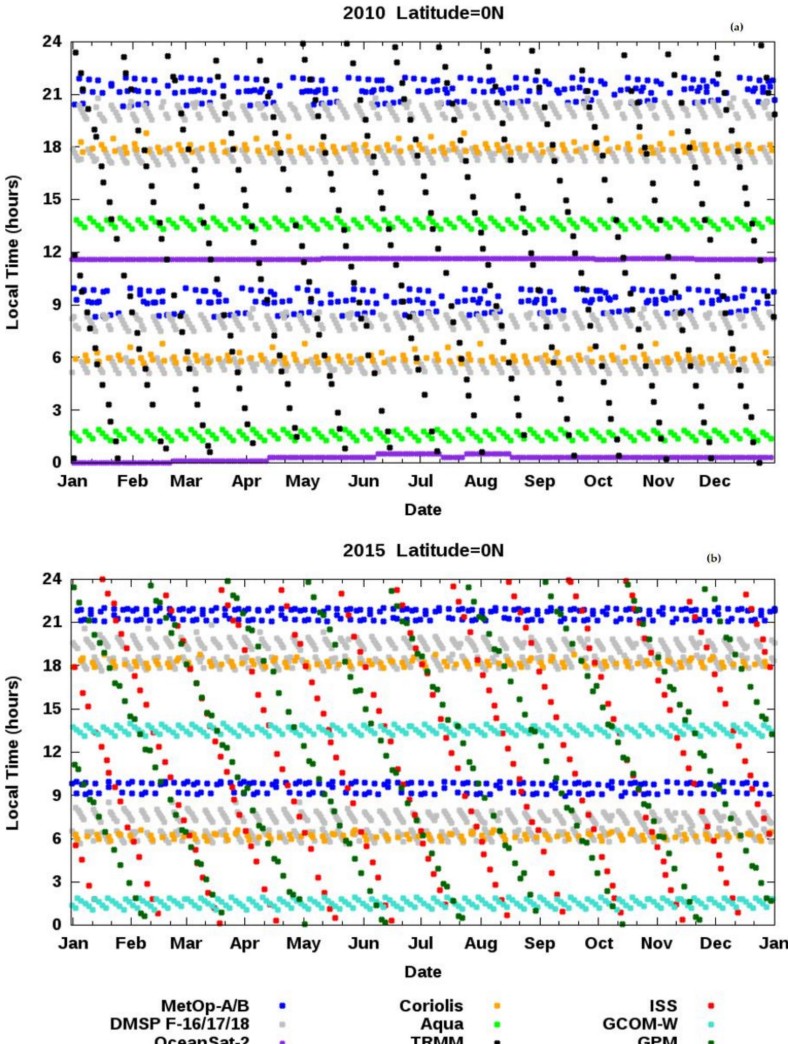

**Figure 2.** Time series of the local time of observation for a location at the equator, for the collection of wind radiometers and scatterometers (Table 1) available during 2010 (**a**) and 2015 (**b**).

If *n* = 3 or 4, the desired diurnal coefficients $(a_0, a_1, a_2)$ and $(b_0, b_1, b_2)$ can be defined such that

$$
\begin{pmatrix} u_1 \\ v_1 \\ \vdots \\ u_n \\ v_n \end{pmatrix} = \begin{pmatrix} a_0 + a_1\cos(2\pi t_1/24) + a_2\sin(2\pi t_1/24) \\ b_0 + b_1\cos(2\pi t_1/24) + b_2\sin(2\pi t_1/24) \\ \vdots \\ a_0 + a_1\cos(2\pi t_n/24) + a_2\sin(2\pi t_n/24) \\ b_0 + b_1\cos(2\pi t_n/24) + b_2\sin(2\pi t_n/24) \end{pmatrix}
\tag{1}
$$

where $(t_1, t_2, \ldots . t_n)$ are the local times in hours of the *n* observations, and equality is only required up to the uncertainty in each of the instantaneous wind estimates. Defining $U = (u_1, u_2, \ldots . u_n)^T$ and $V = (v_1, v_2, \ldots . v)^T$ and separating the unknowns into $\vec{x} = (a_0, a_1, a_2)^T$ and $\vec{y} = (b_0, b_1, b_2)^T$, then (1) can be rewritten as a system of two separate equations, $\vec{U} = [A]\vec{x}$ and $\vec{V} = [A]\vec{y}$, where

$$
[A] = \begin{pmatrix} 1 & \cos(2\pi t_1/24) & \sin(2\pi t_1/24) \\ \vdots & \vdots & \vdots \\ 1 & \cos(2\pi t_n/24) & \sin(2\pi t_n/24) \end{pmatrix}
\tag{2}
$$

If $n = 5$ or $6$, then the system of equations can be expanded such that $\vec{x} = (a_0\ a_1\ a_2\ a_3\ a_4)^T$ and $\vec{y} = (b_0\ b_1\ b_2\ b_3\ b_4)^T$ and $[A]$ becomes a $n \times 5$ matrix,

$$[A] = \begin{pmatrix} 1 & \cos(2\pi t_1/24) & \sin(2\pi t_1/24) & \cos(4\pi t_1/24) & \sin(4\pi t_1/24) \\ \vdots & \vdots & & \vdots \\ 1 & \cos(2\pi t_n/24) & \sin(2\pi t_n/24) & \cos(4\pi t_n/24) & \sin(4\pi t_n/24) \end{pmatrix} \tag{3}$$

In either case, the solution can be written in matrix form as

$$\vec{x} = \left(A^T D_u^{-1} A\right)^{-1} A^T D_u^{-1} \vec{U} \tag{4a}$$

$$\vec{y} = \left(A^T D_v^{-1} A\right)^{-1} A^T D_v^{-1} \vec{V} \tag{4b}$$

where $D_u$ and $D_v$ are the diagonal $n \times n$ matrix carrying the variance of the $u$ and $v$ measurements, respectively (when $n = 3$ or $5$, $[A]$ is square and (4) reduces to $\vec{x} = [A]^{-1}\vec{U}$ and $\vec{y} = [A]^{-1}\vec{V}$). Gille et al. (2005) used this type of formulation during the 2003 QuikSCAT/Seawinds tandem period ($n = 4$, so Equation (1)) was slightly overdetermined) to study the sea breeze diurnal variability near coasts. In their case, all data from this period were mapped onto a 0.25-degree grid and individually solved for each gridpoint, assuming $D_u$ and $D_v$ as identity matrices.

The addition of a wind-speed-only (denoted by $w$) sensor observation from a passive MW radiometer brings in an additional observation at times $t_{n+1}$. For example, during 2010 (Figure 2), the MW radiometer onboard the Aqua satellite filled in the time gap near 13 UTC, and occasionally the asynchronous TRMM observations would occur near 15 UTC, both filling in missing time gaps. Note that in 2015, after TRMM had ended, both RapidScat and GPM provided asynchronous observations. Since the relation between $w$ and $(u, v)$ is nonlinear, the simplest way to incorporate $w$ and avoid nonlinearities is to turn it into a hypothetical vector observation by discretely sampling all possible wind directions $\theta = (0, 1, 2, \ldots 359)$ one degree at a time and, for each case, create the hypothetical observation at time $n + 1$, e.g., for $n = 3$ or $4$,

$$u_{n+1} = w\cos\theta = a_0 + a_1\cos(2\pi t_{n+1}/24) + a_2\sin(2\pi t_{n+1}/24) \tag{5a}$$

$$v_{n+1} = w\sin\theta = b_0 + b_1\cos(2\pi t_{n+1}/24) + b_2\sin(2\pi t_{n+1}/24) \tag{5b}$$

or for $n = 5$ or $6$,

$$u_{n+1} = w\cos\theta = a_0 + a_1\cos(2\pi t_{n+1}/24) + a_2\sin(2\pi t_{n+1}/24) + a_3\cos(4\pi t_{n+1}/24) + a_4\sin(4\pi t_{n+1}/24) \tag{6a}$$

$$v_{n+1} = w\cos\theta = b_0 + b_1\cos(2\pi t_{n+1}/24) + b_2\sin(2\pi t_{n+1}/24) + b_3\cos(4\pi t_{n+1}/24) + b_4\sin(4\pi t_{n+1}/24) \tag{6b}$$

For each value of $\theta$, the residual error $E(\theta)$ is calculated and compared with the observations, e.g., for the case of $n = 4$,

$$E(\theta) = \sum_{i=1}^{4} (a_0 + a_1\cos(2\pi t_i/24) + a_2\sin(2\pi t_i/24) - u_i)^2 + (b_0 + b_1\cos(2\pi t_i/24) + b_2\sin(2\pi t_i/24) - v_i)^2 \tag{7}$$

where $I = 4$ corresponds to the additional windspeed observation, and then retain the solution for $\theta$ which minimizes $E(\theta)$ in (7). This simple minimization procedure locates $\theta$ to the nearest degree, sufficient accuracy for this purpose. This formulation can be expanded to bring in additional wind-speed-only observations at other times, $n + 2$, $n + 3$, etc. In reality, the solution for $\theta$ should not physically deviate too far from nearby time (e.g., within 3-h) scatterometer-derived wind directions. In this investigation, $\theta$ was not allowed to exceed by more than 60 degrees relative to the direction of the wind direction from either

of the preceding or following observations. This reduces the computational burden for the number of search directions when two or more radiometers are included.

The above procedure will only produce meaningful conditions for analysis where the wind vector and wind speed observations are sufficiently numerous and sampled fairly equally across the day. For example, if a TRMM overpass time occurs within one hour of a QuikSCAT observation, it does not contribute much to the total daily sampling, and any bias between the wind speeds could introduce unrealistic high-amplitude semidiurnal components. To assign realistic values to the $D_u$ and $D_v$ diagonal covariance matrices in (4), the variance of the $u$ and $v$ observations ($\sigma_u$ and $\sigma_u$, respectively) within a surrounding 1-degree box (discussed below) was computed for each of the satellite data types used.

It is important to note that the solution of (4) does not assure evidence of physical diurnal wind variability. The magnitude of the diurnal $u$ and $v$ components should sufficiently exceed the variability owing to the variability in the estimated regression coefficients. Defining the peak estimated magnitudes as $A_u = \left(a_1^2 + a_2^2\right)^{1/2}$ and $A_v = \left(b_1^2 + b_2^2\right)^{1/2}$, and assuming that $(a_1, a_2, b_1, b_2)$ are each independently distributed Gaussian random variables, a reasonable acceptance condition is that $A_u$ and $A_v$ both exceed the magnitude that results when the regression coefficients (4) are varied by up to two standard deviations about their mean value, in order to identify conditions where the estimated diurnal wind variability is statistically meaningful. For cases that pass this criterion after solving for (4), an additional coefficient of determination ($R^2$ test) is applied, and cases where $R^2 < 0.95$ are rejected.

Figure 3 demonstrates this analysis for a single day, using satellite observations within a 1-degree region just eastward of the Yucatan peninsula on 16 May 2016, an area of known coastal wind variability. In this example, there are five wind vector observations (two each from ASCAT and WindSat, one from RapidScat, all shown with red symbols) and two wind speed observations (two from AMSR-2, blue symbols), fairly well-spaced through the day. The symbols in the top two panels show the observed zonal and meridional winds, and the middle row shows the wind speed magnitude and direction. After solving for $\theta$ by locating the two values of which minimize (7), the assigned AMSR-2 wind directions are near 275 and 310 degrees (towards the west, onshore), shown in the right middle panel. The least-squares fitted diurnal (blue line) and semidiurnal (red line) wind components are shown in the top two panels. The actual $(u, v)$ and $w$ observations fall on top of the corresponding regression fits (solid black lines). The wind speed is fairly steady during the morning hours and decreases in the afternoon, consistent with the sea breeze conditions typical of this region. While the wind direction does not change markedly, the net effect of including the smaller semidiurnal component is to "flatten" the magnitude of the sea breeze during the morning hours. From this analysis, the "sense" of the elliptical variability can be determined; here, the diurnal (blue) and a smaller semidiurnal (red) wind ellipse spin clockwise (CW) with magnitudes $< 3$ m s$^{-1}$.

Since the analysis in the following section will compare the satellite-derived daily wind components to other observations over multiple years, this analysis should be able to replicate daily wind patterns that are known to regularly occur. Repeating this procedure for each day between 2003 and 2017, where the observations meet the acceptance criteria, Figure 4 shows the regression-fitted diurnal $u$ (top) and $v$ (below) wind components, for a different 1-degree region off the coast of Colombia. Each line corresponds to one day during this interval (the line color serves only to separate the lines for better visual appearance). The predominant maximum in the $u$ and $v$ components is evident near 18Z and 6Z, respectively.

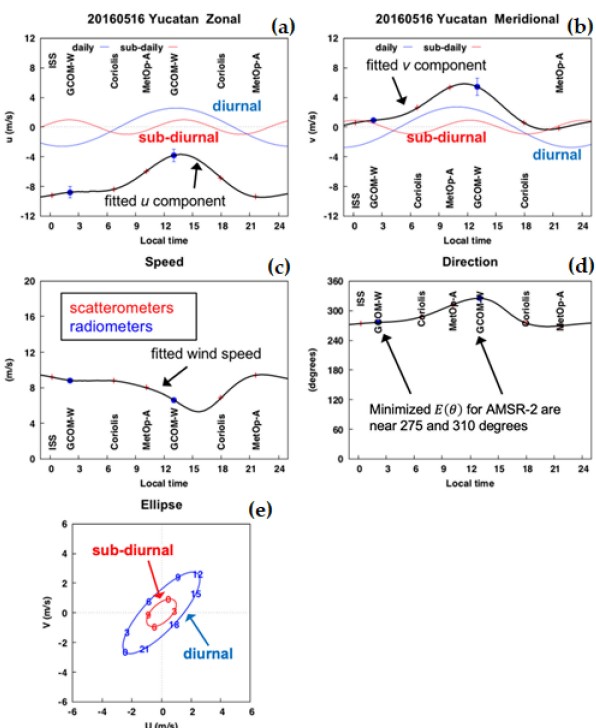

**Figure 3.** Diurnal wind mode analysis for one day (16 May 2016) of observations, offshore of the Yucatan peninsula. On this day, the data consisted of RapidScat, two ASCAT, two WindSat and two AMSR-2 observations (*n* = 7). Each "clump" of points represents the collection of 25-km wind vector cells that falls into this region. (**a,b**) The estimated diurnal and semidiurnal *(u, v)* wind components in blue and red, respectively, and the "fitted" (*u, v*) components. (**c,d**) Same data as in the top panels, but shown in speed and direction (*w, θ*) coordinates. (**e**) Wind ellipses formed by the (mean-subtracted) diurnal and semidiurnal wind components. Both the diurnal (blue) and a smaller semidiurnal (red) wind ellipse spin clockwise with magnitudes < 3 m s$^{-1}$.

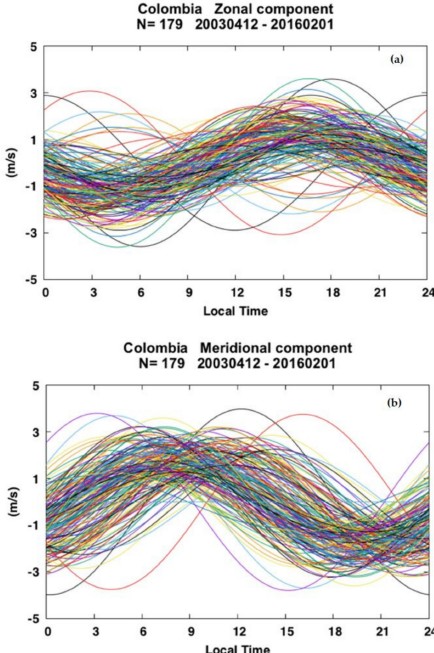

**Figure 4.** Diurnal wind components estimated from analysis of all data (179 cases) between 12 April 2003 and 1 February 2016, for a location just offshore of Colombia. (**a**). Zonal (*u*) component. (**b**) Meridional (*v*) component.

Figure 5 shows a map of the maximum amplitude of the diurnal (*u, v*) wind components, for all observations meeting the quality control criteria during the RapidScat operations period (October 2014 to September 2016). The strong near-coastal diurnal wind patterns (red areas near coastal areas, including the areas from Figures 3 and 4) agree well with the locations determined by [20]. Across the open ocean tropical latitudes, the diurnal components are smaller, less than 1 m s$^{-1}$. For example, there are regions in the ITCZ (box outlining the domain of the TAO-TRITON moored array) where the meridional diurnal wind component appears to be larger than the zonal diurnal component. In the next section, the analysis will be focused within the tropical latitudes, specifically within the open tropical basins (anchored by the TAO-TRITON, RAMA and PIRATA mooring locations), where offshore propagation from coastal effects is diminished [11].

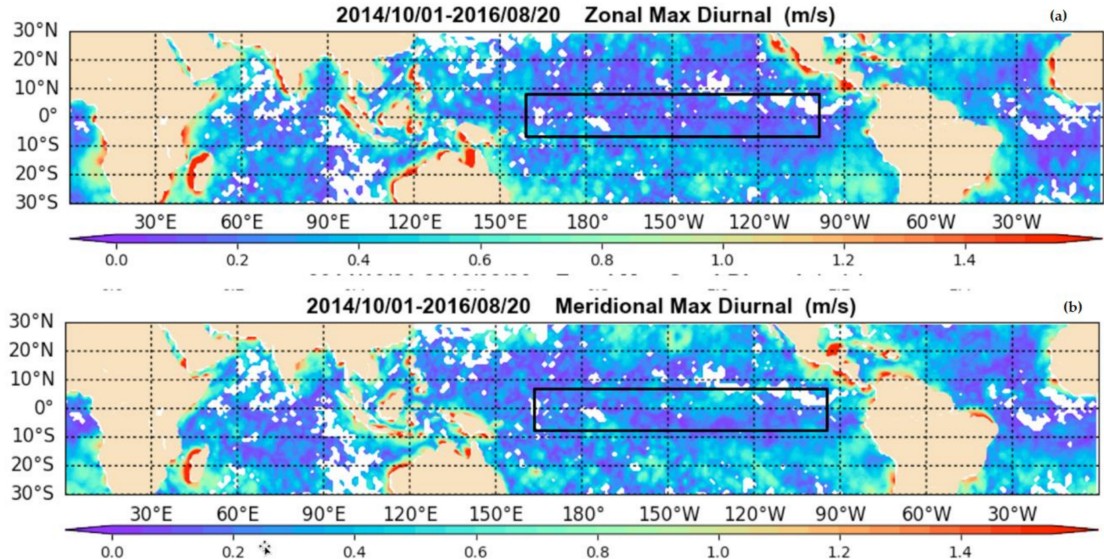

**Figure 5.** Maps of the maximum amplitude of the diurnal wind components, for all observations meeting the quality control criteria during the RapidScat operations period (October 2014 to September 2016). (**a**) Zonal (*u*) component. (**b**) Meridional (*v*) component.

## 3. Daily Wind Modes over the Tropical Pacific

Less understood are wind variability effects in the tropical oceans, far away from coastal effects. While the factors associated with daily variability in wind modes are likely ocean-basin-dependent, these patterns have been surmised to occur owing to variations in the sea level pressure gradient associated with atmospheric thermal tides [31], or potentially due to diurnal modes in the wind vector derivative fields (e.g., surface wind convergence) associated with tropical convection [32]. To date, the best long-term record of surface observations in the tropical Pacific comes from the TAO-TRITON mooring array. Using these data, the zonal wind variability has previously been shown to be dominated by the semidiurnal component, and the meridional wind variability by the diurnal component [9,10]. Figure 6 shows the diurnal wind components (upper left = zonal, upper right = meridional) as revealed with all TAO-TRITON data during 2015 for the mooring located at 2S 155W, in the same panel layout used for Figure 3. The elliptical variability assumption assumed in (1) very closely replicates these observations. In this case, the predominant wind blows from the east at around 5 m s$^{-1}$ (middle panels), but buried within this value is the diurnal wind variability, which has a magnitude of only around 0.2 m s$^{-1}$, which is nearly an order of magnitude smaller than the strong near-coastal wind variability noted in Figure 3. Note that the diurnal mode wind ellipse spins CW, and the semidiurnal spins counter-CW.

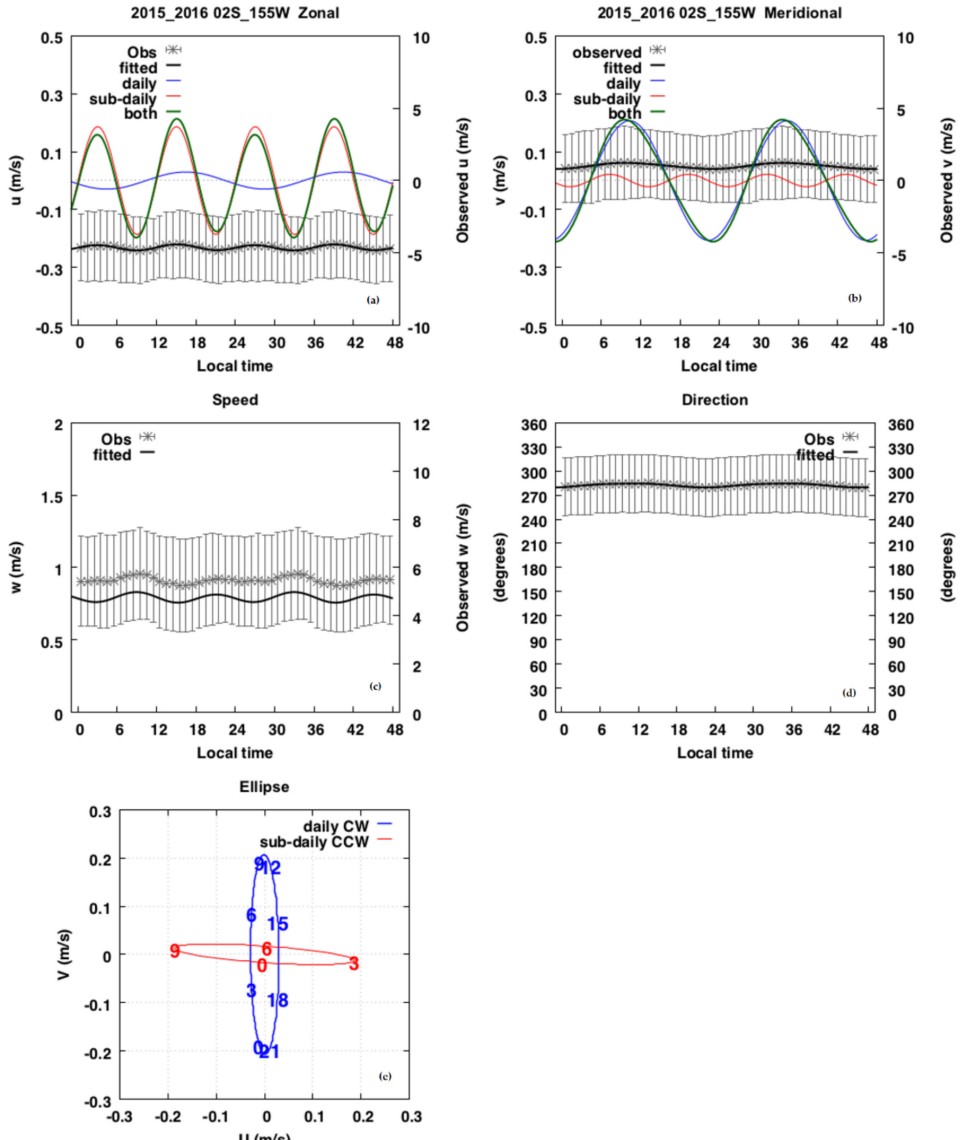

**Figure 6.** Same layout as Figure 3, except for the TAO-TRITON mooring located at 2S 155W. All hourly data from 2015 are used to solve for the coefficients in Equation (4). The "fitted" mean value at each hour is shown by the connected black line, with the associated $\pm 1$ standard deviation bar. The semidiurnal and diurnal modes are evident for the zonal and meridional wind components. The diurnal (blue) and a semidiurnal (red) wind ellipse spin clockwise and counter-clockwise, respectively, with magnitudes $< 0.2$ m s$^{-1}$.

From satellite-based data, the use of scatterometry has not been fully explored to see if it can replicate these known tropical diurnal wind patterns. If so, this would potentially expand the use of scatterometer wind data to serve as an independent validation of the diurnal winds produced by climate models. In this section, regions of diurnal and semidiurnal wind variability are investigated over the tropical Pacific Ocean (8S-8N, 170E-110W) covered by the TAO-TRITON array (boxed area in Figure 5). After the SeaWinds/QuikSCAT tandem period ended in late 2003, the next opportunity for two or more scatterometers came in 2007, with the availability of the first Advanced Scatterometer (ASCAT) on MetOp-A. Therefore, the collection of scatterometers and radiometers from Table 1 are examined during the mid-2007–early 2017 period for individual days when the observations are spaced in time such that all 10 harmonic terms $(a_0\ a_1\ a_2\ a_3\ a_4)$ and $(b_0\ b_1\ b_2\ b_3\ b_4)$ can be estimated with at least two extra degrees of freedom ($n = 12$). Figure 7 shows that over this period of record, this alignment of observations will occur most often if at least

two speed-only radiometers are included. The corresponding number of observations available at each mooring location ranges from 61 (2S 110W location) to 153 (2N 110W location) for the 2007–2017 period. Interestingly, when the two radiometer observations are restricted to only TMI and GMI (preferred asynchronous orbiting), the number of cases is less than if all radiometers except TMI and GMI are used. This is likely due to the Aqua and GCOM-W observing times filling in an otherwise lengthy revisit in the local afternoon times (Figure 2).

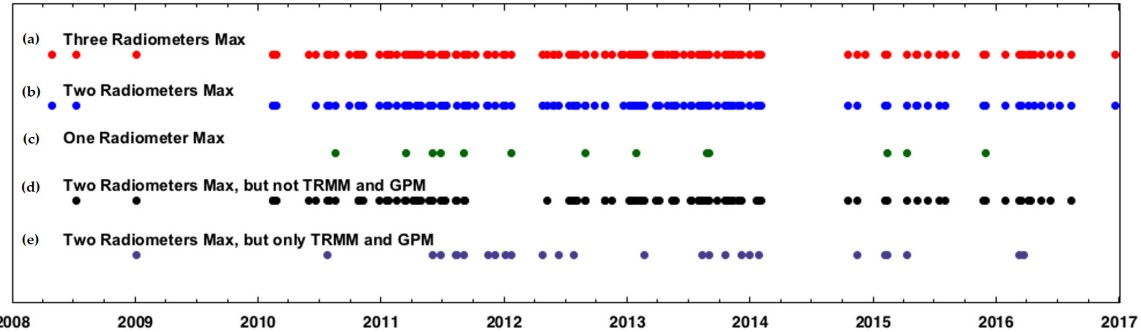

**Figure 7.** Time sequence of days where there are at least 12 *u* and *v* observations surrounding the TAO mooring located at 2S 125W, sufficient to solve Equation (4) with at least two extra degrees of freedom. Each of the five horizontal lines corresponds to a different satellite data acceptance criterion. From top to bottom: (a) All available wind vector observations and up to three radiometers maximum; (b) Same as (a) but two radiometers maximum; (c) Same as (a) but one radiometer maximum; (d) Same as (b) but not allowing TRMM or GPM radiometers; (e) Same as (b) but only allowing TRMM and GPM. The gap in 2014 occurs in the period after OceanSat-2 had ceased functioning but RapidScat had not yet been deployed.

In the discussion to follow, the results of the merged scatterometer + radiometer (denoted as SCAT + RAD) wind mode analysis will be carried out in 1-degree boxes centered at the TAO mooring locations. For graphical purposes, only three latitudes between (2S, 0S and 2N) and eight longitudes from 165E to 95W (24 locations) within the equatorial subset of the TAO-TRITON array will be presented.

### 3.1. Average Daily Wind Modes at the TAO-TRITON Locations

The merged SCAT + RAD wind mode analysis is accompanied by the wind mode analysis carried out with the 3-hourly (i.e., 8 samples per day, sufficient to resolve the diurnal and semidiurnal modes) MERRA2 and ERA-I data, over this same time period. As discussed in the Introduction, the ability of a model to represent the daily modes of variability of wind (and precipitation) provides a test of the physical convective parameterizations used in the model [15,16]. It is recognized that there are known differences in how these and other models partition wind kinetic energy into zonal and meridional wind components, which has implications for surface wind derivative fields such as divergence [33]. Here, the purpose of including these data is to examine how the smaller wind mode variability from two well-known forecast model reanalyses replicates, or deviates from, the patterns of diurnal wind mode variability noted from the TAO-TRITON observations and the SCAT + RAD merging analysis.

Figure 8 summarizes the results of the wind mode analysis in 1-degree boxes centered at the mooring locations, for the zonal (*u*) diurnal + semidiurnal (sum of both) wind component. All figure panels use the same scale, with local time on the abscissa (extending over two daily cycles for clarity, 0–48-h local time) and wind speed on the ordinate between $\pm 1$ m s$^{-1}$. The daily mean zonal wind component (i.e., the $a_o$ and $b_o$ terms in Equation (1)) has been removed to isolate the magnitude of the diurnal variability. The estimates from the TAO moorings, ERA-I, MERRA2 and the SCAT + RAD analysis are represented by the red, blue, green and black line colors, respectively.

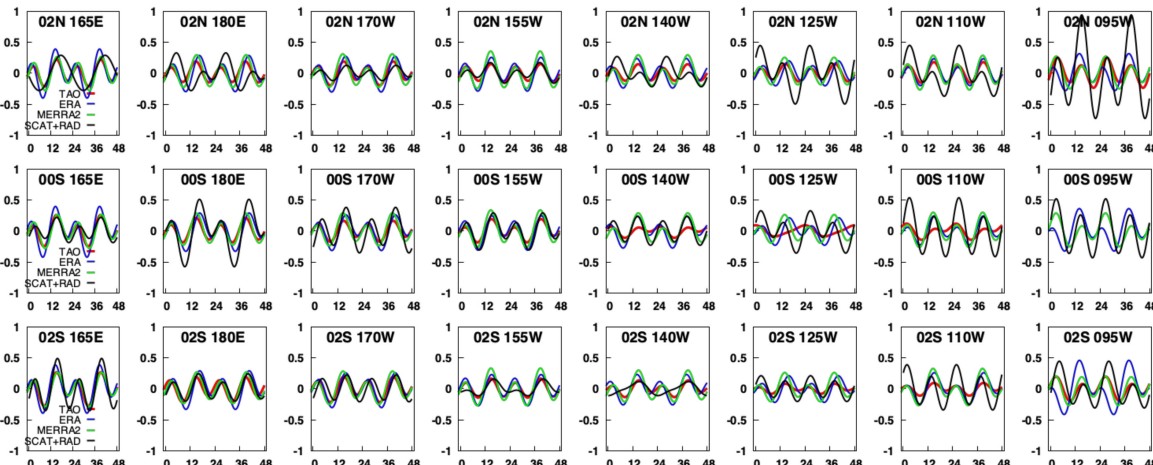

**Figure 8.** Map layout of the daily mode (sum of diurnal + semidiurnal components) zonal (*u*) wind component, at each of the 24 locations within the TAO-TRITON array (box area shown in Figure 5) using all data between 2007 and 2017. The *x*-axis extends from 0 to 48 h local time (two daily cycles) for enhanced visibility, and the *y*-axis extends between ±1 m s$^{-1}$. Positive (negative) values indicate westerly (easterly) winds relative to the daily mean. The estimates from the TAO moorings, ERA-I, MERRA2 and the scatterometer + radiometer (SCAT + RAD) analysis are represented by the red, blue, green and black line colors, respectively.

The TAO-TRITON zonal component shows a predominant semidiurnal variability under ±0.5 m s$^{-1}$, especially for longitudes at and westward of 155W (toward the West Pacific warm pool area), with some added diurnal variability (e.g., 0S 125W location) especially at locations eastward of 155W, in general agreement with [9]. The peak zonal amplitude is fairly consistent, occurring twice daily near 3- and 15-h local time. The semidiurnal zonal wind behavior is believed to be associated with known semidiurnal variations in the tropical sea level pressure gradient, associated with atmospheric thermal tides [31]. The SCAT + RAD zonal component is fairly well aligned in phasing and amplitude with the moorings at and westward of 155W longitude. Eastward of this, the SCAT + RAD component exhibits more added diurnal variability than the moorings and with a two-times larger overall amplitude (nearly ±1 m s$^{-1}$ at the 2N 95W location). By contrast, the zonal component revealed by both models is more consistent with the TAO-TRITON analysis. However, it is noted that the models differ more from each other in the locations where SCAT + RAD also differs from TAO-TRITON (e.g., 2N 95W) and where TAO-TRITON reveals added diurnal variability (e.g., 0S 125W).

Consistent with the layout of Figure 8, the results of the wind mode analysis for the meridional (*v*) wind component are shown in Figure 9. The TAO-TRITON data reveal a predominant diurnal variability, slightly larger in peak-to-peak amplitude than the zonal component, in general agreement with [9]. Moving eastward in longitude (left to right in the figure panels), the maximum in the meridional wind component shifts from the early morning (6 h) more towards midday (12 local). The SCAT + RAD analysis also reveals a diurnal variability, but with substantial additional semidiurnal variability in most of the locations and up to two-times larger overall amplitude relative to the moorings. The patterns in the meridional component revealed by both models are less consistent. In general, the meridional wind component shows generally less semidiurnal behavior (e.g., near 110W), although some locations suggest a superposition of diurnal and semidiurnal patterns (e.g., near 140W). In general, the TAO-TRITON meridional component is smaller in the southwestern locations compared to the northwest locations. Overall, a key takeaway is that, relative to these surface observations, the SCAT + RAD meridional mode does not exhibit as clear a dominant diurnal signal as noted in the moorings and models.

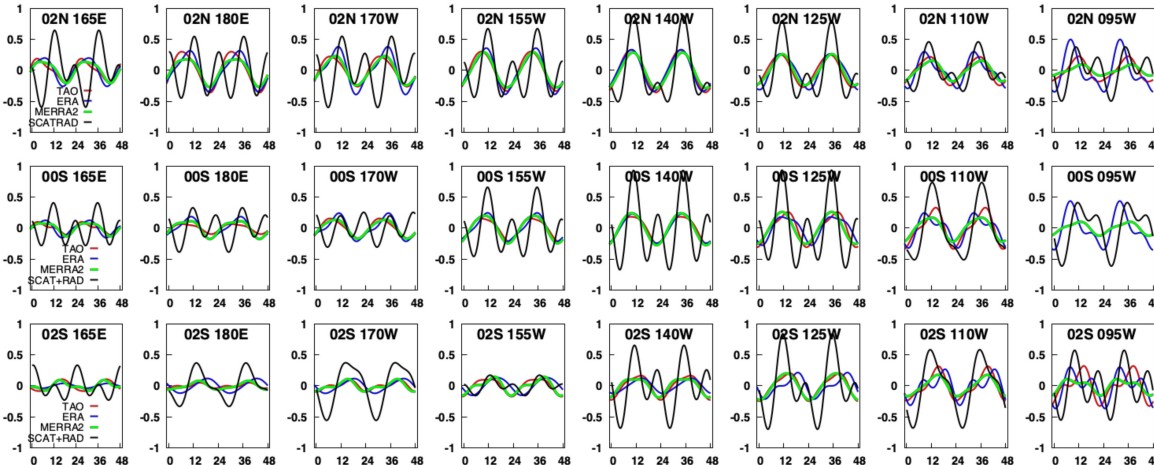

**Figure 9.** Same as Figure 8, but for meridional (*v*) wind component. Positive (negative) values indicate southerly (northerly) winds relative to the daily mean. The estimates from the TAO moorings, ERA-I, MERRA2 and the scatterometer + radiometer (SCAT + RAD) analysis are represented by the red, blue, green and black line colors, respectively.

The overall 11-year (2007–2017) analysis presented above averages any year-to-year variability that may be present in the underlying wind conditions. As noted, such a long-term average was necessary in order to maximize SCAT + RAD data sufficiency at each of the locations. From Figure 7, the SCAT + RAD sampling was densest in the 2010–2014 period when OceanSat-2 and MetOp-A were jointly operating, with a notable gap until RapidScat began operation in late 2014. In the following section, these same data are analyzed for variability as a function of El Niño Southern Oscillation (ENSO) conditions inferred from the Oceanic Niño Index (ONI) [34]. The ONI is based on the variation of the 3-month running mean of the oceanic sea surface temperature (SST) within the domain specified by the 5S–5N, 170W–120W latitude and longitude range, respectively. It is emphasized that the limited and intermittent satellite data over any single year are insufficient to analyze the characteristics of the SCAT + RAD wind variability at many of the TAO-TRITON locations.

### 3.2. Daily Wind Modes from 2010 and 2015

The ITCZ normally moves northward (southward) during northern (southern) hemisphere summer seasons. During El Niño years, the location of the ITCZ shifts further southward with the change in the predominantly zonal trade winds, with resultant changes to the regions of SST and resultant convective precipitation. It is hypothesized that any change to the diurnal wind variability that is manifested as a result of these large-scale trade wind and SST conditions is most likely to be noticeable when El Niño and La Niña conditions are contrasted.

During the 2007–2017 average period presented above, 2007 and 2010 were strong La Niña years, and 2011 was a moderate La Niña year. Moreover, 2009 was a moderate El Niño year, and 2015 was a very strong El Niño year. The remainder were more or less neutral. While presenting and interpreting results for each of these 11 years is beyond the scope and length of this manuscript, in this section, the zonal and meridional wind variability during the 2010 La Niña year is contrasted with the variability during the very strong 2015 El Niño year.

#### 3.2.1. Zonal Wind Modes

Figure 10 depicts the zonal (*u*) wind component estimated by TAO-TRITON, ERA-I, MERRA2 and SCAT + RAD using data from 2010. Figure 11 depicts the same using data from 2015. In both figures, the layout is identical to Figure 8. To ensure that the full year variability was sampled, at least 10 months of TAO-TRITON sampling was required for

each location; otherwise, these data (solid red line) are not shown. As mentioned above, many locations have no SCAT + RAD data and are not shown (solid black line).

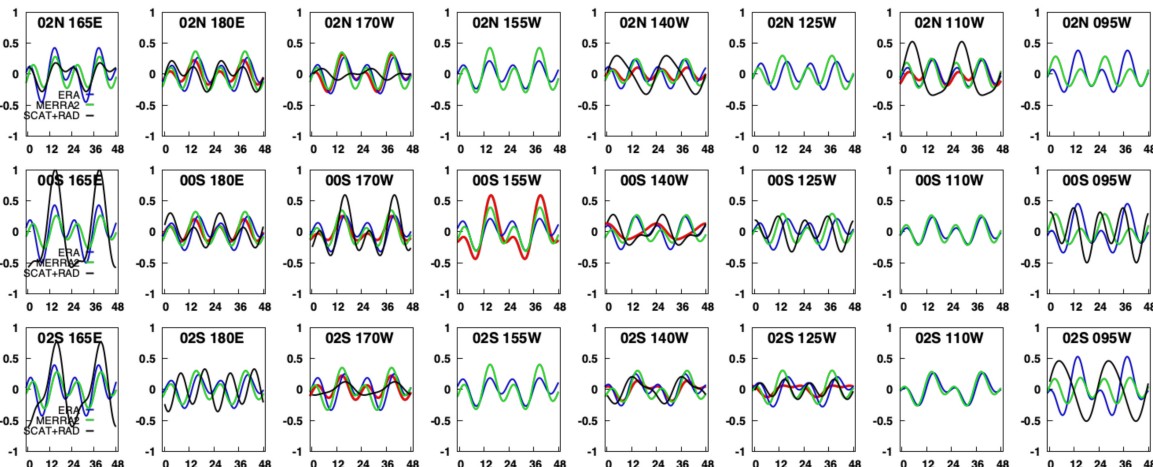

**Figure 10.** Same as Figure 8, but showing zonal (*u*) wind components from 2010, a year with moderate La Niña conditions. Positive (negative) values indicate westerly (easterly) winds relative to the daily mean. For many locations, TAO-TRITON and SCAT + RAD data are not shown owing to the limited number of observations.

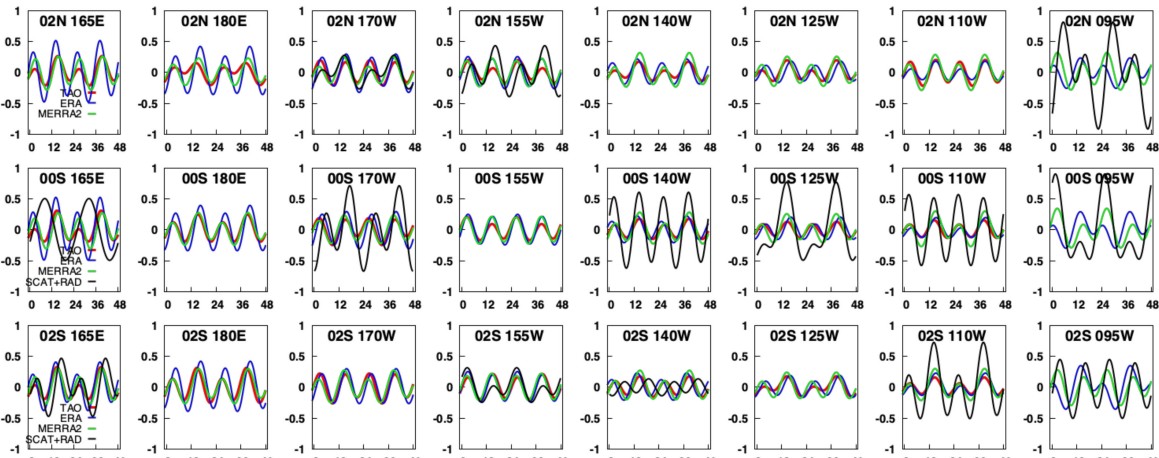

**Figure 11.** Same as Figure 8, but showing zonal (*u*) wind components from 2015, a year with strong El Niño conditions. Positive (negative) values indicate westerly (easterly) winds relative to the daily mean. For many locations, the TAO-TRITON and SCAT + RAD data are not shown owing to the limited number of observations.

While there are some slight differences between 2010 and 2015 data, it is difficult to discern different characteristics in the predominant semidiurnal zonal wind mode between the two models. ERA-I exhibits larger amplitudes compared to MERRA2, in the westernmost (165E longitude) and easternmost (95W longitude) TAO-TRITON locations. On the easternmost side, both reanalysis models tend to introduce a small diurnal component on top of the semidiurnal mode, especially for ERA-I. Compared to 2010, the SCAT + RAD zonal component (where available) during 2015 is slightly larger in magnitude than the TAO-TRITON data. In general, the models and the SCAT + RAD data are more in-phase (i.e., synchronized in local time maxima) with the TAO-TRITON data in 2015 than they are in 2010.

### 3.2.2. Meridional Wind Modes

Figure 12 depicts the meridional (*v*) wind component estimated by TAO-TRITON, ERA-I, MERRA2 and SCAT + RAD using data from 2010. Figure 13 depicts the same using data from 2015. In both figures, the layout is identical to Figure 9.

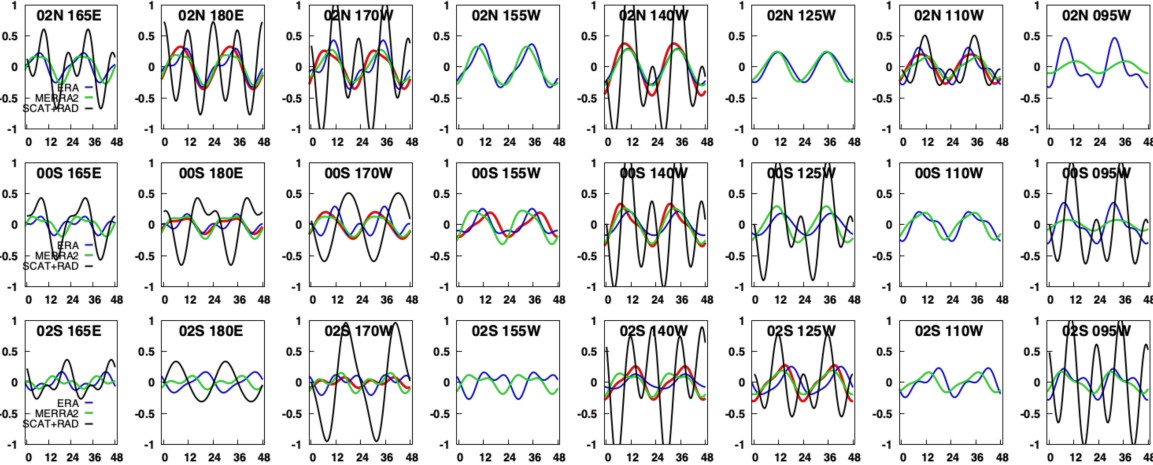

**Figure 12.** Same as Figure 9, but showing meridional (*v*) wind components from 2010, a year with moderate La Niña conditions. Positive (negative) values indicate southerly (northerly) winds relative to the daily mean. For many locations, the TAO-TRITON and SCAT + RAD data are not shown owing to the limited number of observations.

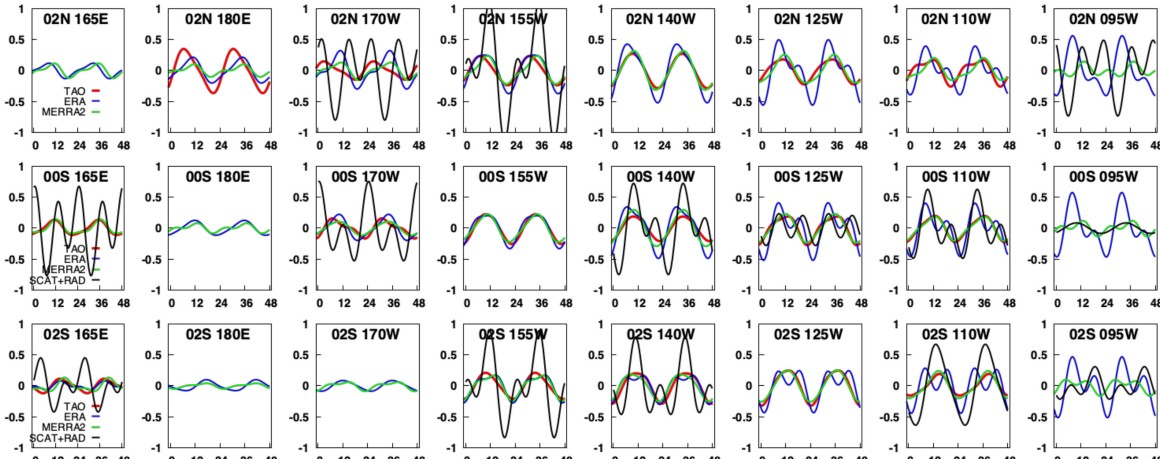

**Figure 13.** Same as Figure 9, but showing meridional (*v*) wind components from 2015, a year with strong El Niño conditions. Positive (negative) values indicate southerly (northerly) winds relative to the daily mean. For many locations, the TAO-TRITON and SCAT + RAD data are not shown owing to the limited number of observations.

In general, the meridional components exhibit more variability amongst the datasets than was noted in the zonal wind analysis. Compared to the zonal component, there is more variability between the two models across mooring locations, especially at the easternmost (95W longitude) locations. Across the southern (2S) longitudes, both models tend to introduce more sub-diurnal variability during 2010 than for 2015. The diurnal mode in the meridional component is largest in the Central Pacific (northeast corner of these 24 panels) and is noticeably smaller in the Western Pacific (southwest corner).

The conclusions from this analysis are that there are noticeable differences in the meridional wind components between these two models when separated by La Niña and El Niño conditions. However, there is also high regional variability, shown by the differences from location to location, which are discussed further in Section 4.

When SCAT + RAD data are available at a location for 2010 and 2015, these data show large meridional wind component differences. However, it is cautioned against drawing conclusions from these data. There are many locations where the behavior of the SCAT + RAD wind mode shows a fairly large sub-diurnal component (e.g., 2N 140W in Figure 12). This could happen when a speed-only (radiometer) observation occurs very close in local time to a vector (scatterometer) observation, and the radiometer wind speed is biased high relative to the scatterometer. The minimization (Equation (7)) will converge by fitting a high-amplitude second harmonic (sub-daily) term to the observations from that particular day. In order to mitigate this condition somewhat, radiometer observations are rejected when they are within one hour of a scatterometer observation, which works well to filter out these cases for the full 2007–2017 meridional analysis shown in Figure 9. However, for this yearly analysis, the smaller data count limits the scientific interpretation of the year-by-year SCAT + RAD data.

## 4. Discussion

In both the SCAT + RAD wind modes and in the MERRA2 and ERA-I data, the zonal wind component is dominated by semidiurnal variability, as demonstrated and explained in earlier studies of TAO-TRITON data by [9]. The interesting results from this study appear in the meridional wind component, for which TAO-TRITON shows mostly diurnal wind variability, but regional and seasonal differences reveal mixtures of diurnal and semidiurnal modes. Both the MERRA2 and the SCAT + RAD analysis exhibited more variability in the meridional diurnal wind component than the TAO-TRITON mooring data. This feature is also noted in the TAO-TRITON data themselves, so it is not clear from these findings if there is a more physical reason for the meridional component, or whether it is related to characteristics of the satellite and mooring measurements.

Nevertheless, these differences in the meridional component suggest a more careful analysis of the ocean wind data compared to the diurnal variability noted in oceanic precipitation, available from the lengthy TRMM and GPM data record. More specific to linking winds and convective precipitation are the dynamically important quantities of time-mean surface convergence/divergence and vorticity, which can be computed from the $(u, v)$ wind components estimated by wind-vector-capable scatterometers [14]. Relative to the wind field itself, the wind vorticity is highly intermittent and more variable [35] and, when estimated from a scatterometer, affected by rain effects. Analysis of QuikSCAT scatterometer data near the ITCZ has revealed the Eastern and Central Pacific Ocean to be characterized by stronger surface convergence than the Western Pacific warm pool and the Indian Ocean [36].

The ITCZ essentially tracks the ascending branch of the Hadley cell, producing convection by solar heating, whereas the opposite occurs on the descending branch. The descending branch is dominated by a subtropical high-pressure area, which suppresses convection. The Southern Pacific Converge Zone (SPCZ) extends southeast from the tropical West Pacific warm pool toward French Polynesia, although its size and orientation vary [37], especially during strong El Niño years. For comparison, Figure 14 contrasts the February GPM IMERG precipitation total from the IMERG data during winter 2015 (strong El Niño) and winter 2017 (weak La Niña). Within the ITCZ, the precipitation patterns appear markedly different, especially noted in 2015, where the SPCZ moves northward and further east, into the regions covered by the TAO-TRITON moorings. Within the ITCZ latitudes, the analysis of GPM-IMERG satellite precipitation data generally follows a diurnal pattern, with a maximum near 5 UTC, but the diurnal amplitude does not appear to be noticeably different from year to year (figure not shown). More coordinated investigations of these joint data, including sampling of the TAO-TRITON mooring data to match the satellite sampling times [38], is needed before further connections between tropical convective processes and oceanic wind variability can be concluded.

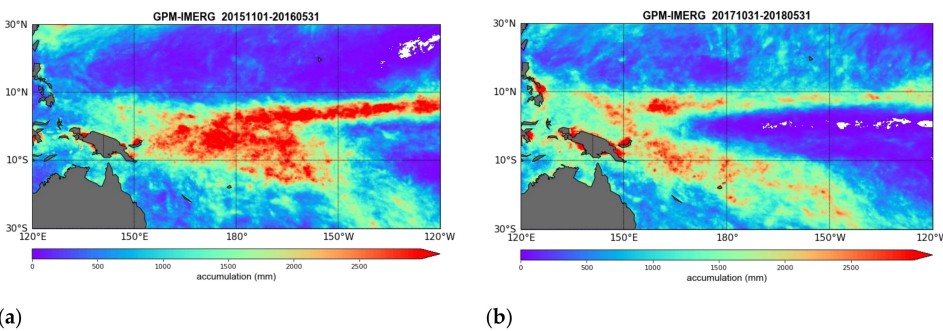

**Figure 14.** Comparison of GPM IMERG 7-month accumulated precipitation (mm) during November 2015–June 2016 (**a**) and November 2017–June 2018 (**b**). Approximate domains of the RAMA, TAO/TRITON and PIRATA moored arrays are shown. The color scale extends from 0 to 3000 mm.

## 5. Conclusions

Since the relationship between the ocean wind speed (*w*) and its zonal (*u*) and meridional (*v*) wind components is nonlinear, a methodology has been demonstrated to merge the speed-only measurements from passive MW radiometers with the wind vector measurements available from scatterometers, to estimate the daily modes (diurnal and semidiurnal) of wind vector component variability. Compared to TAO-TRITON mooring array data, the SCAT + RAD wind components and the MERRA2 and ERA-I reanalysis model wind components fairly well replicate the semidiurnal zonal wind variability over the tropical Pacific (in agreement with earlier studies) but generally show more variability in the predominantly diurnal meridional wind components. Compared to the two models, the SCAT + RAD meridional component differs widely across the mooring locations or shows no clear dominant diurnal or semidiurnal mode. It is hypothesized that the discrepancies amongst the meridional winds are due to interactions between surface convergence and convective precipitation over tropical ocean basins.

Looking ahead, few studies have attempted to examine the wind derivative fields (e.g., convergence, vorticity) that provide linkage to oceanic convection (e.g., [32]). This is due in part to the intermittent nature of available ocean vector winds and precipitation radar coverage. Scatterometer vector winds from operational, sun-synchronous orbiting satellites provide practically the only observational means to estimate these dynamically important wind derivative fields. Beginning in 2017, ocean wind speed observations from CYGNSS provide additional diurnal wind sampling. Currently, the Compact Ocean Wind Vector Radiometer (COVWR) is scheduled to begin operations from the ISS in 2021. These data combine with the operational C-band (ASCAT) and Ku-band scatterometer data (e.g., ScatSat, FY-3) to continue the record for examining the daily modes of wind vector variability [39]. For example, does the predominance of strong convergence lead to convective systems that shift or alter the modes of the diurnal wind cycle? Further use of these data together with GPM Dual-Frequency Precipitation Radar (DPR) observations is encouraged, to identify and highlight the mechanisms linking tropical convection processes.

**Author Contributions:** Conceptualization, F.J.T., S.H.-V.; methodology, F.J.T.; software, F.J.T.; validation, F.J.T., S.H.-V., D.G.; formal analysis, F.J.T.; investigation, F.J.T., D.G.; resources, F.J.T.; data curation, F.J.T.; writing—original draft preparation, F.J.T.; writing—review and editing, F.J.T., S.H.-V., D.G.; visualization, F.J.T.; supervision, F.J.T.; project administration, F.J.T.; funding acquisition, F.J.T. All authors have read and agreed to the published version of the manuscript.

**Funding:** The authors acknowledge support from the NASA Ocean Vector Winds Science Team via program element NNH13ZDA001N-OVWST.

**Institutional Review Board Statement:** Not applicable.

**Informed Consent Statement:** Not applicable.

**Data Availability Statement:** Not applicable.

**Acknowledgments:** The authors acknowledge Ziad Haddad who suggested the satellite merging methodology. The discussions with (and related work carried out by) Sarah Gille and Thomas Kilpatrick contributed to the explanation of the results. Mark Bourassa is acknowledged for his leadership of the NASA Ocean Vector Winds science team. The authors acknowledge the many scientific contributions by W. Timothy Liu (1947–2020) in bridging the areas of tropical meteorology and oceanography. This work was carried out at the Jet Propulsion Laboratory, California Institute of Technology, under a contract with the National Aeronautics and Space Administration. Copyright 2020. All rights reserved.

**Conflicts of Interest:** The authors declare no conflict of interest. The funders had no role in the design of the study; in the collection, analyses, or interpretation of data; in the writing of the manuscript, or in the decision to publish the results.

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
