# Peer review of "Examination of the Daily Cycle Wind Vector Modes of Variability from the Constellation of Microwave Scatterometers and Radiometers"

_remotesensing, doi:10.3390/rs13010141_

Round 1
Reviewer 1 Report
My main concern is that to my opinion the accuracy of the satellite method is not made clear enough. Maybe this has to do with the presentation of the results, see the detailed points below.
(1) Figures 8 to 15 are not well readable, notably the labels along the axes. This makes it difficult to compare the figures. I recommend the authors to change the layout. Have they considered 8 rows of 3 panels instead of 3 rows with 8 panels?
(2) Figures 8 and 9 show an average over the years, while in figures 10 to 15 the years are split. Why? This makes it more difficult to intercompare the results and to properly assess the merits of the satellite method. Why not average the results of the various methods over the common time period and present them in one panel?
(3) Caption of figure 8: it is not clear from the caption that this figure contains the satellite results.
(4) Satellite data are sparse over the daily cycle, while buoy measurements and NWP data have a better coverage. Have the authors considered replacing the satellite measurements by simultaneous buoy measurements or NWP data, calculate the diurnal and semi diurnal parameters, and compare these with the original ones in order to assess the accuracy of the satellite method compared to more traditional methods?
Reviewer 2 Report
Review of Turk et al.
This paper tests a new methodology for combining radiometer-based wind speeds with scatterometer-based vector winds in order to better resolve the diurnal cycle of ocean winds, and compares the results with buoy networks and model reanalyses. The methodology, which is new but still grounded in the literature, is promising and the comparison with the buoys/models is thorough. My main concern with the paper is the visualization of the data. Subplots within key figures are extremely small and show a lot of data each. This makes it difficult for the reader to understand at an intuitive level how all the different datasets actually compare. I am going to recommend major changes to the figures, but since the authors don’t need to change their data/analysis, the effort likely won’t take them that long and thus overall we are talking about minor revisions.
- Figures 8-15 require substantial revision. They all show diurnal cycles of wind components at various locations, from the authors’ radiometer/scatterometer analysis, from buoys, and from model reanalyses. The authors discuss in the text how the different datasets compare, but since the radiometer/scatterometer combination is in one set of figures, the buoys in another, and the models in yet another, and each of the figures is comprised of more than a dozen tiny subplots, it is incredibly difficult for the reader to intuitively understand how the datasets actually compare. To a first approximation, the authors’ discussion has to be taken on faith because the figures do not facilitate interpretation that would support the discussion. There are other additional complicating factors as well, such as Figs. 8-9 having different ordinate scales than the other figures.
I propose that the authors rethink how they present their data in these figures. My overarching suggestion is that everything be made larger, the text more legible, and that there be fewer curves per dataset per subplot. One idea I have is to rotate the base figure to landscape orientation (enabling the entire image to take up a whole page), and plot in each subplot one curve for the radiometer/scatterometer analysis, one curve for the buoys, and one curve for each of the reanalyses. That way, every curve uses the same ordinate and it is much easier to understand at a glance how the different datasets compare at each location.
The authors likely could summarize their entire results shown in Figs. 8-15 with just two figures that way (one per wind component) and it would be much easier to understand the authors’ discussion points. However, one concern the authors may have is how to best represent the interannual variability. I might suggest using shading to do that (e.g., a mean curve with shading representing the uncertainty based on interannual variability). If the authors feel compelled to discuss certain years (e.g., El Nino/La Nina comparison), they could do so using additional figures, since my suggested approach would free up a lot of figure space (by condensing 8 figures to 2).
I am certainly open to the authors coming up with a different approach, but the current status of Figs. 8-15 is that they unfortunately add very little to the discussion. The paper needs a simpler way of comparing the different datasets. It can’t be done easily with different datasets in different figures.
- Figure 1 – The authors claim no significant bias, but it does seem as though the right tails of the distributions are different. There are not as many data points there, so I agree the bias effect is likely small. However, the authors should quantify what the mean/median bias is in these subplots.
- One other thing about Fig. 8 and 9 – the current captions don’t explicitly identify these data as coming from the radiometer/scatterometer analysis. Please change this.
- The CCMP product, which I am sure the authors are aware of, is not really discussed in this paper despite its obvious relevance to it. I understand not wanting to add more analysis of another dataset, but since CCMP does a qualitatively similar combination of scatterometer and radiometer winds, and provides data in a 6-hourly format that at least coarsely resolves the diurnal cycle, the authors really need to discuss that dataset and explain why they didn’t use it. For example, what advantages does the authors’ technique provide over CCMP?
